# PolySona: Parameter-Efficient and Modular Latent Behavior Modeling for Traffic Simulation

## Abstract

In rare but safety-critical driving scenarios, we hypothesize that trajectory outcomes become increasingly multi-modal based on differences between driver style compared to non-critical, common scenarios. However, current approaches for trajectory prediction rarely account for differences in driving style, which may lead to "averaged" driving style in predictions. While average-case behavior may work well in straight driving, easy scenarios, it limits the diversity of outcomes in more complex scenes or in rare events. Extraction of driving style has several benefits, as it enables simulation of counterfactual outcomes in real-world log replays and potentially more accurate predictions through style-consistent predictions. In this paper, we present a parameter-efficient Mixture-of-Experts framework for extraction of latent driving styles in trajectory prediction models. We choose a parameter-efficient approach to reduce forgetting in well-generalized trajectory prediction models, while offering portability of trained driving style modules. We also propose a *Style Consistency Metric* to quantify how often a model's multi-modal outputs cover the true driving style. In our results, we benchmark different mixture-of-LoRA approaches with our method and show qualitative results that show how the learned experts specialize, and how model saliency changes with our approach. Additional qualitative results can be found on our project website: https://missanonybloon.github.io/polysona/

## 1 Introduction

As autonomous driving becomes an increasingly accessible technology, handling *mixed autonomy* traffic systems will also become an increasingly important research question. Mixed autonomy traffic systems comprise of both human drivers and autonomous drivers, to varying degrees. This may already the case in large cities, where autonomous driving has scaled to commercial use (and limited personal use). The general autonomous driving stack approaches driving in sequential modules, each responsible for a specific task: perception, prediction, planning, and control. Prediction, in particular, is responsible for predicting traffic states *that have not occurred yet*. Much of the difficulty in trajectory prediction lies not in the common cases such as lane following and sparser suburban roads, but rather the complex cases with many stressors involved. With more stressors, human driving behavior diverges into a splay of different outcomes; this is expected due to the General Adaptation Syndrome, a spectrum of "fight or flight" responses in humans which has also been shown to influence crowd navigation behavior of humans (Kim et al., 2012).

Currently, this variation in human decision making of other human drivers on the road has not been modeled in trajectory prediction frameworks of autonomous vehicles. Yet, this variable would influence multi-modal outcomes in cases, where an arbitrary decision produces drastically different trajectories, even with a fixed traffic context and route intent. Errors from trajectory prediction can trickle and accumulate down to planning and control modules, resulting in safety implications, especially in risky scenarios. We hypothesize that *driving style* is increasingly impactful during these rare events, where there are typically more stressors pressuring drivers to make decisions in a split second. One motivating example is the case where the driver needs to change lanes to achieve their route goals, but is directly blocked by other drivers in the target lane. If the driver changes lane without modifying their current speed, they would collide with the other drivers. This condition is

Figure 1: **Motivating Example: Lane change occupied by other vehicles.** The ego vehicle has the goal of changing lanes and reaching a particular waypoint. At its current velocity, a direct lane change would result in a collision with the other vehicles. Here, the ego vehicle is faced with a natural stressor to make a decision: either decelerate and merge behind the other vehicles, or accelerate and pass the other vehicles first, then merge. Both outcomes are plausible, yet lead to drastically different trajectories by average displacement error metrics, which are computed per-timestep.

an example of a stressor on the road. The driver must then choose between two specific outcomes: decelerate and change lanes behind the other drivers, or accelerate and change lanes in front of other drivers. In terms of common trajectory prediction metrics, the difference between the two very plausible outcomes is large; one decision would be penalized, while the other's likelihood would be increased with respect to model parameters. We illustrate this example in Figure 1, where the green vehicle's possible trajectory paths diverge greatly per timestep.

Our approach is to model latent driving style as a parameter-efficient Mixture-of-Experts, where we train several expert Low-Rank Adapters (LoRA) (Hu et al., 2022) guided by real world priors on driving style. To guide learned experts towards representations related to driving style, we use a vehicle traffic dynamics adaptation to the Social Forces Model (SFM) for expert routing. This approach is inspired by the use of SFM for other learning-based robotics tasks, such as drone planning (Pang et al., 2021) for collision avoidance among multiple moving agents (i.e. cars) while navigating towards the goal. We motivate the use of parameter-efficient paradigms for its efficiency and baseline performance guarantees. Additionally, adapters make it easy to control for expert behavior, which opens possibilities for counterfactual simulation outcomes.

Our main contributions are summarized as follows:

1. A parameter-efficient and simple Mixture-of-LoRA approach for latent modeling of driving style in trajectory prediction (Fig. 2).
2. A router design guided by social force features from traffic scenarios;
3. A style miss rate metric to benchmark simple style consistencies between ground truth trajectories and predictions;
4. Qualitative analysis on expert specialization with our MoL approach for latent variable modeling.

## 2 RELATED WORKS

### 2.1 DRIVING STYLE MODELING IN AUTONOMOUS DRIVING

Driving style can be useful for several modules in the autonomous driving stack. For policy training, style modeling can be used for personalization of driving policies to maximize rider comfort, especially if the policy is expected to mimic the human's own driving (Karagulle et al., 2024; Schrum et al., 2024). On the other hand, driving style can also be useful for planning, where accurately predicted human driving behavior is essential. Alternative simulated outcomes can also be useful for benchmarking planning modules against counterfactual human behavior. In trajectory forecasting, some work has investigated driving styles for specific maneuvers, such as lane changes (Liu et al., 2021; Hao et al., 2024), or specific context such as highways (Xing et al., 2020) and intersection (Wang et al., 2023) scenarios. Driving style is well-motivated as it can describe variation in plausible outcomes when context and intent are fixed. In our work, context and intent are fixed via the base model; we model latent variable given the sample context and the selected intents from the base model.

Several efforts have been made to model driving style from observable trajectory features as a means to extract driving style quantitatively, such as with Graph Neural Networks (GNNs) (Chen et al., 2022; Chandra et al., 2020) or classical clustering approaches based on kinematic properties of trajectories (Hao et al., 2024; Xing et al., 2020). Most recent efforts model driving style with deep learning, using either recurrent networks (Liu et al., 2021; Xing et al., 2020; Choi et al., 2021) or generative models (Kim et al., 2021; Jiao et al., 2022) to extract driving style features implicitly. One common theme of previous work is that driving style is generally considered a latent variable which can be modeled as a function of kinematic properties such as acceleration and jerk. However, many

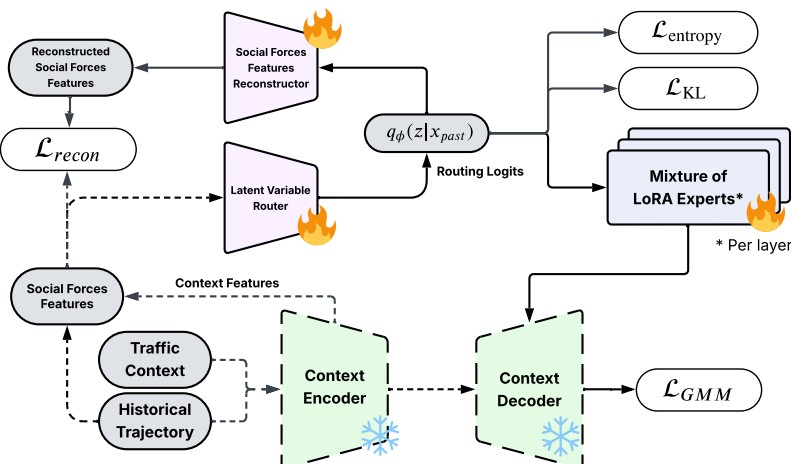

Figure 2: **Training Overview.** We use a Mixture-of-LoRA (MoL) approach to learn a driving style latent variable for trajectory prediction models. Solid lines indicate gradient flow in backpropagation. In summary, we construct social forces features from historical agent trajectories and train a routing network in a Variational Autoencoder-like fashion; in other words, the router predicts logits for discrete latent classes, and a reconstructor network then learns to re-construct social forces features. The predicted routing logits and then used to select experts in the MoL layers, which are attached to certain decoder layers of the base model. Experts are combined using the weight mixing scheme illustrated in Figure 3. All modules except the base model are trained end-to-end.

of these works require an annotated dataset (Liu et al., 2021) or a specialized backbone architecture to extract driving style (Kim et al., 2021; Chandra et al., 2020). The most relevant recent work in controllable behavior modeling with latent variables may be TAE (Jiao et al., 2022), which also uses a VAE-like objective, but models the driving style latent as a continuous instead of categorical variable. There are several limitations of this work. Firstly, the latent variable being continuous makes it difficult to interpret the latent. Secondly, TAE does not achieve competetive results to SOTA trajectory prediction frameworks, which limits the realism of simulated outcomes. We show this comparison in our experimental results to emphasize the performance gap in previous work to our proposed method in realistic prediction.

In our work, we distinguish our goals from previous works in that we want to complement and maintain performance from state-of-the-art trajectory prediction models, whilst also being able to extract generalized categorical representations of driving style.

## 2.2 PARAMETER-EFFICIENT MIXTURE OF EXPERTS FOR SPECIALIZATION IN DRIVING

Parameter-efficient Mixture-of-Experts (PE-MoE) is a key focus in language modeling and image diffusion, where efficient fine-tuning atop foundation models is essential. Accordingly, prior PE-MoE work has largely centered on language and diffusion tasks. We extend these paradigms to trajectory prediction, aiming to extract a latent variable that captures driving style. PE-MoEs typically use collections of LoRA adapters as experts. Single-LoRA adapters can encode specialized concepts in image diffusion (Gandikota et al., 2024; Gu et al., 2023), while in language tasks, PE-MoEs mitigate catastrophic forgetting (Dou et al., 2023) and match the performance of full fine-tuning at lower cost (Zadouri et al., 2024; Li & Zhou, 2025). Using multiple pre-trained LoRAs also enables reusing task-specific skills. MoE approaches in this context aim to train a robust router to allocate experts per sample. An survey of adapter merging techniques was recently published by Yadav et al.Yadav et al. (2024). Multi-task learning with LoRAs similarly targets a latent skill representation (task-skill matrix), reusing skills across tasks, akin to PE-MoEs. Notable examples include PolytroponPonti et al. (2023), C-Poly (Wang et al., 2024a), and Hyperformer (Karimi Mahabadi et al., 2021). These are complementary to our work, and our MoE design is compatible with most, with some limitations on the differentiability of global expert routers, which we will discuss in the next section.

Some works apply parameter-efficient fine-tuning approaches to trajectory prediction. Forecast-PEFT (Wang et al., 2024b) uses prompt vectors and LoRA layers to improve trajectory prediction performance in downstream tasks and achieve close results to full fine-tuning, similarly to language

models. On the other hand, Munir et al. explore the possibility of using Large Language Models in trajectory prediction (Munir et al., 2024), thus their application of LoRAs would be in the language domain. Our use case is different; we learn a latent variable across different driving styles in a parameter-efficient manner. In robotics, MoEs have been applied to vehicle trajectory prediction, often focusing on goal or intention inference (Yuan et al., 2024). We instead focus on driving style, with the intention of maintaining driving style continuity between past trajectories and future predictions. MoEs have also improved drone trajectory prediction (Fraser et al., 2023). Alternative latent-variable methods like VAEs (Xu et al., 2022) are more computationally costly and limit latent portability across models.

## 3 METHODOLOGY

### 3.1 LATENT DRIVING STYLES WITH EXPERTS

Driving styles are difficult to model because the notion of "driving style" is ill-defined and non-standardized, especially in autonomous driving research. Due to the difficulties of labeling driving style consistency and reliably, many datasets often do not include driving style related information, making this problem challenging—akin to latent variable modeling. In this case, we observe *driving style* as an outcome of latent variable modeling.

In our approach, we make the key assumption that driving style must be strongly correlated to *social forces between agents*. Second-order kinematics is directly related to the actions a driver takes with the steering wheel, throttle, and brake, as input to control a vehicle. That is, the actions patterns that a driver takes on steering, throttle, and brake should be directly correlated with the driving style. The distribution of social forces on each agent over time may be a strong signal related to car-following parameters such as comfortable following distance, preferred velocity, maximum acceleration, and minimum deceleration. Such parameters should not change according to context, and should be constant over time. We illustrate our approach in Figure 2. Our approach first assumes a pre-trained trajectory forecasting model based on encoder-decoder transformer architectures. This model is frozen and serves as a shared representation of general driving behavior. Each MoL layer,

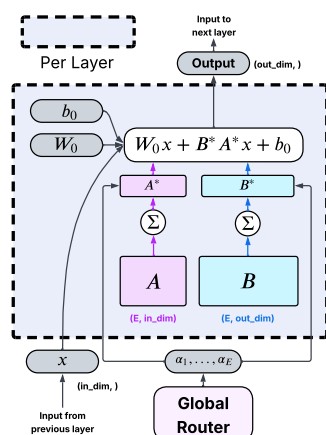

Figure 3: **CAT weight mixing scheme in MoL layers.** In our experiments, we find that the CAT weight-mixing scheme (Prabhakar et al., 2024) performs better on Kalman Hard difficulty scenarios.

depicted in blue, has some variation of the process depicted in Figure 3, which shows the mixing scheme for CAT (Prabhakar et al., 2024), the most recently published method for MoL.

**Mixture of LoRA Experts.** *Low-rank adapters (LoRA)* (Hu et al., 2022) are a popular method for parameter-efficient finetuning (PEFT), which is popularly used for large language models. LoRA represents finetuning as a low-rank update to pre-trained weight matrices $W_0 \in \mathbb{R}^{d \times k}$:

$$h = W_0 x + \Delta W x + b_0 = W_0 x + BA x + b_0 \tag{1}$$

where $B \in \mathbb{R}^{d \times r}$ and $A \in \mathbb{R}^{r \times k}$, and the rank of both matrices $r \ll \min(d, k)$. During training, the pre-trained weights $W_0$ are frozen and only matrices $B$ and $A$ are updated.

*Mixture-of-experts (MoE)* is a modular paradigm which consists of N learned experts $\{E_1, ..., E_N\}$ and a router function which combines outputs from each expert in a weighted voting fashion. The Experts $E_n$ are typically learned jointly and end-to-end, where the objective is for each expert to learn a specialized representation. In the forward pass through a MoE model, the overall output is determined by a weighted "vote" of the individual expert outputs. The voting weights are known as *routing weights*, and a mixture function $G$ determines how the outputs are combined using the routing weights. In our work, $G$ is simply a weighted sum.

For our use case, we would like to learn specialized experts which encode latent driving styles, but also want to avoid the computational burden of full MoEs. Thus, we use Mixture-of-LoRA (MoL),

which trains a set of expert LoRAs, instead of another trajectory prediction model. The mixing paradigm of the LoRA experts can be interchangeable with any existing paradigm for MoL. Thus, we benchmark several in our experiments. We note that MoL has not yet been benchmarked in trajectory prediction so far, to the best of our knowledge. Thus, our experiments are also some of the first benchmarks on applying parameter-efficient techniques to trajectory models.

**Expert Routing based on Social Forces.** We create a global expert router based on social forces and environment context features from the context encoder of the backbone network. The router is treated like a variational posterior estimator $q_\phi(z|x_{past})$ for social forces and context features; we sample the latent differentially with Gumbel-Softmax from the predicted latent logits, then reconstruct the input social forces and context features. We compute social force features from the input trajectory information based on the Social Forces Model. For a particular intersection between vehicles $i$ and $j$, the repulsion force between them is defined as:

$$F_{ij} = \alpha \cdot e^{(\delta - \|x_j - x_i\|)/\beta} \cdot \hat{\theta}_{ij} \qquad (2)$$

where $\alpha$ is a hyperparameter scaling repulsion magnitude, $\beta$ is a hyperparameter scaling the decay rate as pairwise distances become larger, and $\delta$ is a hyperparameter corresponding to preferred, or comfortable, distance between agents. All three hyperparameters are set to 1 in our experiments. When constructing social forces features, interactions between a vehicle and itself are masked out, along with vehicles beyond a radius threshold of 10 meters vehicles not traveling in the same direction. To combine social forces features with context embeddings of hidden dimension $d$, we concatenate both social forces features and context features to a vector of dimension $2d$, then use a single dense layer to fuse features together, resulting in a final feature dimension of $d$. This fused representation is used as input to the global expert router.

The expert router architecture is a simple 2-layer multi-layer perceptron: a single dense layer projecting to a hidden dimension of 64, a normalization layer, ReLU nonlinearity, dropout at p=0.1, and finally a dense layer outputting logits for $E$ classes, or the number of experts. In our experiments, we use three experts ($E = 3$) by following priors from existing traffic psychology literature, which suggests that there are three distinct types of drivers (Klauer et al., 2009). In brief experimentation with higher-complexity global routers, we found that heavier routers overfit quickly and deteriorate performance.

**Loss Objectives.** We use three loss terms to guide the MoL layers towards a driving style representation. Firstly, we maintain the original trajectory prediction objective, which maximizes the likelihood of the ground-truth trajectory given a predicted Gaussian Mixture of trajectory outcomes ($L_{GMM}$). Secondly, since driving style is an unsupervised variable, we enforce a Variational Autoencoder-like (VAE-like) objective on the global router, which eventually serves as a "persona classifier". We have two terms enforced on the router: a reconstruction term, which reconstructs the social forces features $s$ extracted in the previous section, and a Kullback-Leibler divergence term, which maximizes the evidence lower-bound (ELBO), which is a lower bound on the probability of observing the data generated by the model $p_\theta(x)$. Thus, the loss objective in training $\mathcal{L}$ becomes:

$$\mathcal{L} = \mathcal{L}_{\text{GMM}} + \lambda_{\text{recon}}\mathcal{L}_{\text{recon}} + \lambda_{KL}KL(q_\phi(z|x_{\text{past}})||p(z)) + \lambda_{\text{entropy}}\mathcal{L}_{\text{entropy}} \qquad (3)$$

$$\mathcal{L}_{\text{GMM}} = \mathbb{E}_{q_\phi}(z|x_{\text{past}})\left[-log p_\theta(x_{\text{future}}|z)\right] \qquad (4)$$

$$\mathcal{L}_{\text{recon}} = \|s_{\text{orig}} - s_{\text{pred}}\|_2 \qquad (5)$$

$$\mathcal{L}_{\text{entropy}} = -\sum_z q_\phi(z \mid x_{\text{past}}) \log q_\phi(z \mid x_{\text{past}}) \qquad (6)$$

## 3.2 MEASURING STYLE CONSISTENCY

In trajectory prediction tasks, standard metrics such as *Average Displacement Error (ADE)* and *Final Displacement Error (FDE)* measure how closely a predicted path follows the ground truth in Euclidean space. While these metrics capture spatial accuracy, they are agnostic to the underlying kinematic signatures (e.g. rapid acceleration or hard braking) that distinguish different driving styles. To address this gap, we introduce a *style consistency metric*, which evaluates whether a model's multi-modal outputs "cover" the true driving style. The core of this metric is a Gaussian Mixture Model (GMM) that assesses style consistency between the ground truth and its predicted trajectories.

Table 1: **Trajectory Prediction Benchmark Performance Comparisons.** We benchmark different variants of our approach with different parameter-efficient MoL schemes (bottom half) compared to existing benchmark trajectory prediction models (top half). TAE (Jiao et al., 2022) is the most recent work in modeling latent driving style to our proposed method, as it does not require any privileged datasets; as shown, the performance is not competitive to baseline trajectory prediction models. All non-baseline experiments are averaged over three fixed random seed runs. Green cells indicate improvements over TAE. Bolded values indicate best-performance in respective columns. Our method improves upon the baseline model in overall trajectory prediction metrics across all MoL schemes. *The best all-around results are achieved with the MoV approach* (Zadouri et al., 2024), which uses IA3 (Liu et al., 2022) for finetuning, instead of LoRA.

| Method | # Trainable Params | ∇Router | brierFDE↓ | minADE↓ | minFDE↓ | MissRate↓ |
|---|---|---|---|---|---|---|
| Autobot (Girgis et al., 2022) | 1.5M | - | 2.4439 | 0.8892 | 1.7817 | 0.2803 |
| Wayformer (Nayakanti et al., 2022) | 15.2M | - | 2.5747 | 0.9348 | 1.9718 | 0.3574 |
| MTR (Shi et al., 2022) | 65.2M | - | 2.1702 | 0.8645 | 1.7094 | 0.3107 |
| MTR+Actions (Zheng et al., 2024) | 65.2M | - | 2.1605 | 0.8658 | 1.7102 | 0.3208 |
| TAE (Jiao et al., 2022) | 249K | - | 12.0023 | 4.8269 | 12.0023 | 0.9275 |
| PolySona+MTR+Polytropon (Ponti et al., 2023) | 527K | ✗ | 2.1602 | 0.8623 | 1.7037 | 0.3164 |
| PolySona+MTR+C-Poly (Wang et al., 2024a) | 969K | ✗ | 2.1613 | 0.8625 | 1.7048 | 0.3169 |
| PolySona+MTR+HyperFormer (Karimi Mahabadi et al., 2021) | 527K | ✗ | 2.1601 | 0.8623 | 1.7038 | 0.3159 |
| PolySona+MTR+CAT (Prabhakar et al., 2024) | 526K | ✓ | 2.1607 | 0.8624 | 1.7041 | 0.3171 |
| PolySona+MTR+MoV (Zadouri et al., 2024) | 195K | ✓ | **2.1588** | **0.8619** | **1.7016** | **0.3158** |

Following the clustering methodology introduced in (Zheng et al., 2025), we argue that a simpler clustering model can be a more interpretable and robust style evaluator than complex deep latent approaches. Specifically, we fit a two-component GMM to summary statistics of the ground-truth trajectories in the validation set. Each trajectory is represented by a feature vector comprising the maximum absolute acceleration, the variance of acceleration, the variance of speed, and the "gamma" statistic (variance of jerk divided by mean jerk) as defined in (Murphey et al., 2009), omitting mean speed to avoid contextual entanglement. We then assign each candidate trajectory to one of the two mixture components—"normal" or "aggressive." Finally, we define the *Style Miss Rate (SMR)* as the fraction of samples for which none of the predicted trajectories share the same style cluster as the ground truth. A lower SMR indicates better coverage of the driver's true style, complementing traditional displacement-based metrics. For clarity, we summarize the SMR formulation below:

$$\text{SMR} \;=\; \frac{1}{N} \sum_{n=1}^{N} \mathbf{1}\big( \forall\, i \in \{1, \ldots, K\},\; s_{n,i} \neq s_n^* \big) \tag{7}$$

where $s_n^*$ is the ground-truth style for sample $n$, $s_{n,i}$ are its $K$ predicted styles, and $\mathbf{1}(\cdot)$ is the indicator function. More details and exact formulation can be found in the Appendix.

## 4 RESULTS

**Hardware.** Each experiment is trained on 32 GB memory, two AMD EPYC 7352 24-Core Processors, and two NVIDIA RTXA5000 GPUs.

**Experiment setup.** In our experiments, the base trajectory prediction model used is Motion Transformer (MTR), a state-of-the-art, open source, non-autoregressive approach. We train experiments on the Argoverse 2 dataset (Wilson et al., 2021), which consists of 250,000 total scenarios. We use 183,333 samples for training and 22,979 samples for validation. Our experiments were conducted with the UniTraj framework (Feng et al., 2024) for trajectory prediction. In our experiments, the task is to predict the next future 6s trajectories, given 2s of agent history trajectories and road context polylines. Agent trajectories are sampled at 10Hz, for a total of 20 history frames and 60 future frames. Baseline experiments which do not involve our proposed method are all trained based on their original recommended hyperparameter settings. All experiments with our proposed method are trained on 10 epochs and evaluated on the 10th epoch. All experiments with

Table 2: **Style Miss Rate between baseline model (*) and variants of our method.**

| Method | SMR↓ |
|---|---|
| MTR* | 0.6317 |
| TAE | 0.7607 |
| Ours+Polytropon | **0.2198** |
| Ours+C-Poly | 0.2279 |
| Ours+HyperFormer | 0.2248 |
| Ours+CAT | 0.2293 |
| Ours+MoV | 0.2215 |

Table 3: **minADE Comparison by Kalman Difficulty and TDBM Driving Style groups.** We also compare variants of our method on different MoL schemes to baseline models and the most recent latent driving style modeling baseline, TAE, across Kalman difficulty categories and TDBM driving styles (Cheung et al., 2018). In general, we find that *weight mixing provides the greatest improvement to samples with hard Kalman difficulty*. None of the variants improved minADE on "careful" driving samples over baseline models; however, "careful" is also the TDBM driving style assigned to the vast majority of common driving scenarios.

| Method | Kalman Difficulty | | | TDBM Driving Styles | | | |
|---|---|---|---|---|---|---|---|
| | Easy | Medium | Hard | Timid | Careful | Reckless | Threatening |
| Autobot (Girgis et al., 2022) | 0.8399 | 1.2436 | 1.9054 | 0.9186 | 0.9765 | 0.8906 | 0.8434 |
| Wayformer (Nayakanti et al., 2022) | 0.8764 | 1.3440 | 3.0969 | 0.9676 | 0.9484 | 0.9362 | 0.8861 |
| MTR (Shi et al., 2022) | 0.8195 | 1.1736 | 3.0148 | 0.8849 | 0.7718 | 0.8670 | 0.8190 |
| MTR+Actions (Zheng et al., 2024) | 0.8212 | 1.1781 | 2.5331 | 0.8846 | 0.7904 | 0.8687 | 0.8194 |
| TAE (Jiao et al., 2022) | 4.1203 | 9.9024 | 20.3377 | 4.8066 | 3.9524 | 4.4860 | 4.8664 |
| PolySona+MTR+PolyTropon (Zadouri et al., 2024) | 0.8185 | 1.1690 | 2.5585 | 0.8803 | 0.8665 | 0.8653 | 0.8156 |
| PolySona+MTR+C-Poly (Wang et al., 2024a) | 0.8187 | 1.1691 | 2.5474 | 0.8803 | 0.8472 | 0.8655 | 0.8166 |
| PolySona+MTR+HyperFormer (Karimi Mahabadi et al., 2021) | 0.8186 | **1.1673** | 2.5707 | 0.8803 | **0.8471** | 0.8653 | 0.8156 |
| PolySona+MTR+CAT (Prabhakar et al., 2024) | 0.8188 | 1.1678 | **2.4978** | **0.8793** | 0.8477 | 0.8655 | 0.8161 |
| PolySona+MTR+MoV (Zadouri et al., 2024) | **0.8180** | 1.1690 | 2.5502 | 0.8803 | 0.8652 | **0.8649** | **0.8150** |

our proposed method are initialized from a MTR+Actions (Zheng et al., 2024) checkpoint trained on Argoverse 2; thus, there is no need to account for domain shift in fine-tuning. All experiments are trained to learn three experts (or driving styles) with a prior probability of $[0.3, 0.6, 0.1]$ for each expert, respectively, which is inspired from driver distributions published by the NHTSA (Klauer et al., 2009). In the base model, there are six decoder layers, where each decoder layer consists of agent and map attention blocks, as well as a Gaussian mixture prediction head. We apply Rank-4 MoL to the object attention and the Gaussian Mixture prediction head of the 3rd and 6th decoder layers (6 decoder layers total). This produces about 30 MoL layers in MTR experiments. More hyperparameter details for experiment reproduction can be found in the Appendix.

**Metrics.** We use four standard metrics from trajectory prediction to measure model performance:

- **BrierFDE** ($\downarrow$): error (m) of each prediction mode of Gaussian Mixture, weighted by the mixture score. Accounts for both trajectory accuracy and model confidence.
- **minADE** ($\downarrow$): Minimum displacement error (m) from the ground truth trajectory of the predicted trajectory modes, averaged over time. Best-case, per-timestep error.
- **minFDE** ($\downarrow$): Minimum final displacement error (m) between the closest ending position of the predicted trajectory and the ground truth trajectory. Best-case, final-timestep error.
- **Miss Rate** ($\downarrow$): The rate at which minFDE is greater than 2 meters. When the Miss Rate is 1, no predictions ended within a radius of 2 meters of the final position of the ground truth.

## 4.1 COMPARISONS TO OTHER LATENT BEHAVIOR METHODS AND MoL VARIANTS

We compare our method to several baselines in trajectory prediction, including a recent method in modeling latent driving style, TAE (Jiao et al., 2022). Latent variable modeling is common in modern trajectory prediction architectures, but may not always pertain specifically to variations in driving style. For example, regression-based trajectory prediction models such as MTR (Shi et al., 2022) and Wayformer (Nayakanti et al., 2022) use Gaussian Mixture prediction heads to model latent output modes based on route intent. Our work focuses on modeling variations in driving style, where we assume intent is fixed. As mentioned in the related works, other approaches for modeling latent driving style in a similar fashion either (1) require a privileged dataset involving human subjects which cannot be released, or (2) do not fix intents when modeling latents. Regardless, we show comparisons to both SOTA trajectory prediction models (which employ latent intent modeling) and SOTA driving style latent variable modeling (TAE).

In Table 1, we compare the overall performance on Brier FDE, minADE, minFDE, and Miss Rate across different trajectory prediction architectures (top half), recent work on latent driving style modeling (TAE (Jiao et al., 2022)), and MoL approaches (bottom half) when applied in our framework. Each experiment is averaged over three fixed random seed runs, except for the baseline models in the top half. As demonstrated by significant gaps in performance metrics for trajectory prediction, our approach, PolySona, shows much better performance on trajectory prediction metrics thanks to its compatibility with state-of-the-art trajectory prediction architectures. Aside from this, PolySona

is also modular and categorical compared to TAE, enabling more controllable simulation outcomes with counterfactual latents.

C-Poly (Wang et al., 2024a), HyperFormer (Karimi Mahabadi et al., 2021), and Polytropon (Ponti et al., 2023) are parameter-efficient approaches from multi-task learning (MTL) literature, rather than from Mixture-of-LoRA. While MTL is typically distinct from MoL work, these MTL methods are very similar in that they use distinct LoRA matrices per task. However, differently from our use case, they 1) assume that LoRA adapters are pre-trained and 2) assume task classes are a given. In other words, expert assignments under these MTL approaches are hard and non-differentiable. This means that the router cannot be learned end-to-end with the experts. And, since we do not have access to ground-truth driving style annotations, we also cannot pre-train the LoRA experts. For MTL experiments, the router only depends on VAE objectives to choose experts, since gradients from the GMM loss do not propagate back to the router ($\nabla$Routing in Table 1). Table 2 shows SMR performance across variants of our method; we achieve considerable improvement of *at minimum* 63.7% across all methods compared to the MTR and TAE baselines, with the best SMR score being with the Polytropon variant (Ponti et al., 2023). We note that improvement on SMR also scales across methods similarly to other performance metrics in Table 1. In Table 3, we show results for baseline models and variants of our approach across several Kalman Difficulty Feng et al. (2024) and TDBM Cheung et al. (2018) subsets of Argoverse 2. Kalman difficulty, originally introduced in UniTraj Feng et al. (2024), is defined as the magnitude of deviation in meters from ground truth when a linear Kalman filter is used to predict the future trajectory. In TDBM, both traffic state features and user study responses are used to determine a linear feature-behavior mapping which maps relative vehicle features to different driving behavior categories. Overall, all variants of our method improves both the overall performance metrics and across most scenarios detailed in Table 3. Amongst different MoL paradigms, we find that MoV (Zadouri et al., 2024) achieves the best all-around performance boosts. As an additional plus, this also the approach with the lowest number of trainable parameters.

## 4.2 INTERPRETING EXPERT SPECIALIZATIONS

We would like to investigate what behavior each expert specializes in from the learned model. To investigate interpretable differences between experts, we compare the second-order kinematic properties of predicted trajectories by each expert. We predict Argoverse 2 validation trajectories, and plot the mean of the absolute value of acceleration ($m/s^2$), jerk ($m/s^3$), angular acceleration ($\theta/s^2$), and angular jerk ($\theta/s^3$). We plot this comparison in Figure 4a.

To visualize whether there are non-trivial latent assignments by the MoL layers, we also plot the t-SNE features computed using router embeddings, which is visualized in Figure 4b. Our router learns a clear three-class separation among social forces features; we also observe empirically that reconstruction loss converges well, implying that three latent classes are sufficient to reconstruct the social forces features of the target agent.

## 4.3 ABLATION STUDY

We conduct an ablation study on our approach by experimenting with alternative variants with respect to router expressivity, full fine-tuning, and contribution of each objective term. In Table 4, we record the percentage difference for each ablation variant of our full approach, which can be referenced in the last row of Table 1. In general, we find that the *greatest degradation in performance occurs when the reconstruction process or KL divergence is removed* during training. Unsurprisingly, the router experiences expert collapse when KL loss is removed. Expert collapse also occurs when entropy loss on the posterior logits is removed, which is aligned with our expectations. Full fine-tuning ($\sim$65M trainable parameters) performs very similarly to parameter-efficient learning; while minADE improves slightly, we find that all other metrics do degrade slightly.

## 4.4 HOW MANY LATENTS ARE ENOUGH?

We also explore how the number of categorical latents modeled influences performance across trajectory prediction metrics, grouped by difficult categories. In particular, we train several variants our model with the CAT weight mixing, varying only the number of latents modeled. Each variant is trained three

Table 5: minADE by # Latent Classes & Kalman Difficulty Level

| | | minADE↓ | | |
|---|---|---|---|---|
| $K$ | Easy | Medium | Hard | Overall |
| 2 | **0.8108** | 1.1951 | 3.5275 | **0.8589** |
| 3 | 0.8188 | **1.1678** | **2.4978** | 0.8624 |
| 4 | 0.8119 | 1.1990 | 3.5116 | 0.8603 |
| 5 | 0.8120 | 1.1996 | 3.5191 | 0.8605 |
| 6 | 0.8109 | 1.1986 | 3.5729 | 0.8595 |
| 8 | 0.8123 | 1.2027 | 3.4217 | 0.8610 |

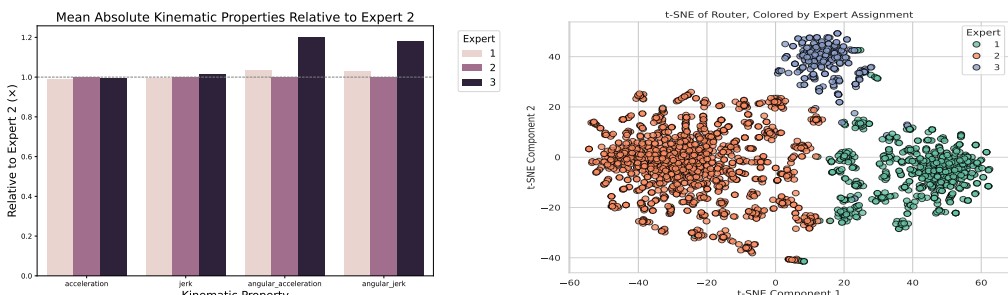

(a) Relative magnitudes of second-order kinematics.      (b) t-SNE visualization of router embeddings.

Figure 4: **Visualizations of expert representations in Ours+CAT.** We plot both relative magnitudes of kinematic properties and projected router embeddings by expert. In (a), we show the mean magnitudes of acceleration, jerk, angular acceleration, and angular jerk between experts, relative to Expert 2. We observe that *the learned latents pertain more to angular, lateral kinematics, as opposed to longitudinal kinematics along the lane.* In (b), we pass the same set of scenarios through the trained router and project the embeddings to 2 dimensions with t-SNE, then color each point by the router's expert prediction. This figure shows that *the learned latents are not collapsing to redundant representations*, which is a common practical concern with latent variable modeling.

Table 4: **Ablation Study.** We evaluate the impact of each component in our MoL training scheme for trajectory prediction using the Ours+CAT variant. All values are averaged over three fixed random seed runs. In this table, we quantify the % improvement in each respective metric. We find that the greatest degradation to performance is from removing the reconstruction process or the KL loss.

| Ours | % brierFDE↑ | % minADE↑ | % minFDE↑ | % MissRate↑ |
|---|---|---|---|---|
| + LinearRouter | -0.153% | 0.024% | -0.226% | 0.157% |
| + Full Finetuning | -0.114% | 0.444% | -0.313% | -0.673% |
| - Social Forces | -0.221% | 0.299% | -0.401% | -1.060% |
| - Context Features | -0.186% | 0.096% | -0.363% | -1.199% |
| - KL Loss | -0.243% | 0.136% | -0.439% | -1.669% |
| - Reconstruction | -0.500% | -0.071% | -0.780% | -1.337% |
| - Expert Entropy Loss | -0.099% | -0.096% | -0.135% | 0.265% |

times across three fixed random seeds and averaged, to account for randomness in results. This experiment is motivated by two questions: 1) How many latents are sufficient to model driving style for difficult scenarios? And 2) As number of latents modeled increase, does the performance also increase due to increased expressivity?

Results are shown in Table 5. As we expect based on our prior knowledge, we observe that minADE performance across Medium and Hard Kalman difficulty scenarios is best when modeling $K = 3$, *especially for hard scenarios*. Despite the overall minADE for $K = 3$ being the highest of all values, we find that this is due to representation being skewed towards Easy scenarios. Still, best overall performance results from modeling $K = 2$, where we find the lowest minADE for Easy scenarios.

To answer the two questions above, our experiment results suggested that *three latent classes is sufficient to model driving style for difficult scenarios*, and that *increasing the number of latents does not necessarily increase overall performance of the model.* Drawing from this, we can also consider traditional VAEs to model an infinite number of latents in a continuous vector space—this result may also illustrate the interpolation of performance to the continuous case.

Interestingly, we find that $K = 3$ also corroborates our prior from from Traffic Psychology literature, where scientists designing human subjects studies initially divided drivers into two "driving style" groups, but later modified the study to account for three "driving style" groups, as having three groups resulted in better separation between varying levels of risk taking (Dingus et al., 2006).

## 5 CONCLUSION

In this paper, we introduced a framework based on Mixture-of-LoRA (MoL) to extract driving style variables from pre-trained trajectory models. The framework is both parameter-efficient and modular, making it easy to adapt to existing trajectory prediction works. This framework allows for controllable routing due to the global router, which learns a latent variable for driving style. Our contributions also offer a metric focused on driving style consistency and qualitative analysis on model saliency, which can be useful for interpreting prediction reasoning in future work.

## 6 REPRODUCIBILITY STATEMENT

We, the authors, emphasize reproducibility for this work, especially since nearly all previous work (to the best of our knowledge) do not have reproducible or open-sourced methodologies. To uphold complete reproducibility of our work, we include an anonymized code link in our supplemental material, in addition to referencing it on our anonymized project site, which can be found at the end of the abstract. Additionally, we include all hyperparameter details for the setup in Figure 2 in Appendix Section 8. Our implementation is based on the unified trajectory prediction framework, UniTraj (Feng et al., 2024), and can thus be integrated with all supported open source architectures and datasets. In our experiments, we primarily use Argoverse 2 (Wilson et al., 2021).

In the future, we plan to open-source our full implementation, including scripts used to run all fixed seed experiments and averaged results. By doing so, we hope to support more research in modeling driving style for autonomous driving.

## 7 USAGE OF LLMs

The use of LLMs in our work is strictly limited to two things: implementation of TAE and aiding in plot styling. While TAE is a relevant comparison for our proposed approach, it is not open sourced; we were unable to obtain the original source implementation. Thus, we re-implemented TAE to the best of our ability using details from the original TAE publication, then used LLMs to refine details on our implementation. All writing was done without the use of LLMs.

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

# APPENDIX

## A  ANONYMIZED CODE

For an anonymized version of our code, please see: anonymous.4open.science/r/polysona

## B  WEIGHT-MIXING IMPLEMENTATION

**PyTorch-Style Forward Pass of Weight-Mixing Mixture-of-LoRA Layer**

```python
expert_alphas = router(...) # (b, num_experts)

expert_weight_A = (expert_alphas[..., None, None] * self.
    expert_weight_A[None]).sum(
    dim=1
)  # (b, r, in_features)
expert_weight_B = (expert_alphas[..., None, None] * self.
    expert_weight_B[None]).sum(
    dim=1
)  # (b, out_features, r)

output = torch.einsum(
    "bi,bri->br", x, expert_weight_A
)  # (b, in_features) @ (b, r, in_features) -> (b, r)
output = torch.einsum(
    "br,bor->bo", output, expert_weight_B
)  # (b, r) @ (b, out_features, r) -> (b, out_features)
output = self.dropout(output)  # (b, out_features)

output = F.linear(x, w0, b0) + output  # w0x + BAx + b0
```

## C  QUALITATIVE RESULTS ON ARGOVERSE

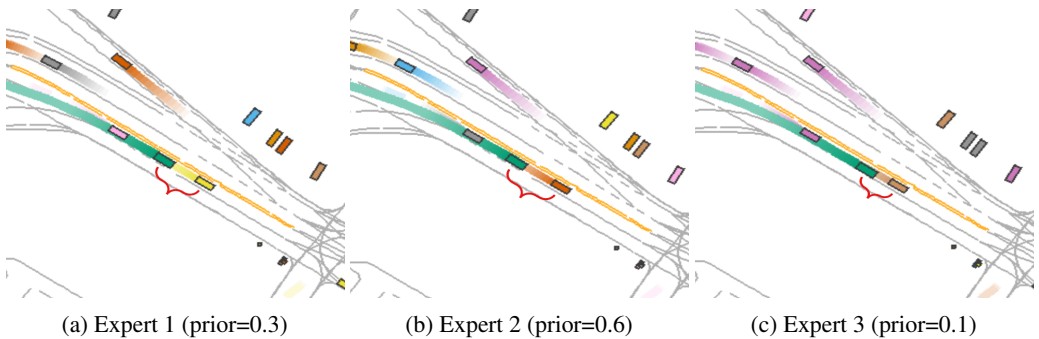

(a) Expert 1 (prior=0.3)    (b) Expert 2 (prior=0.6)    (c) Expert 3 (prior=0.1)

Figure 5: **Qualitative result: Rollouts on a lane change scenario with different experts on Ours+CAT.** We show how different experts behave under the same lane change scenario, where the goal is to lane change between two agents. While rollouts from Expert 1 and 2 are very similar (maintaining at least one vehicle length from the leading vehicle), Expert 3, which has the lowest prior probability, predicts a trajectory with much smaller headway distance. Animations of these scenarios are better visualized on our project website, linked in the abstract.

## D  STYLE CONSISTENCY METRIC VISUALIZATION

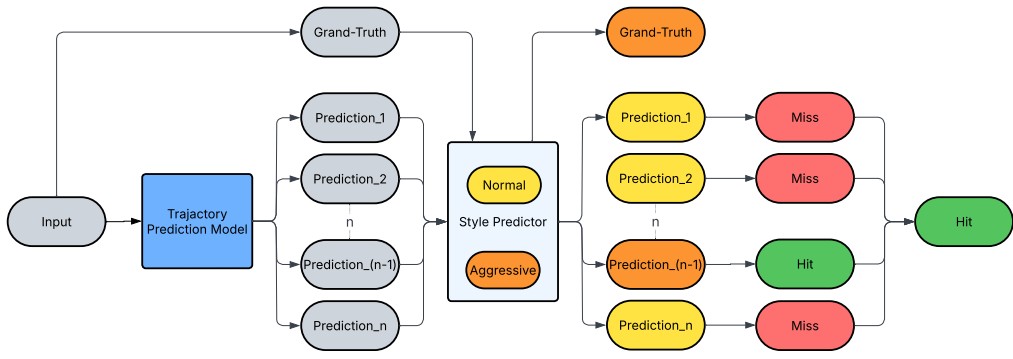

Figure 6: **Style Consistency Metric.** Given a driving scenario ("Input"), the black-box trajectory predictor generates $n$ candidate futures (Prediction$_1$, . . . , Prediction$_n$). A learned style predictor then assigns each candidate—and the true future ("Ground-Truth")—to one of two clusters (e.g. "Normal" vs. "Aggressive"). If at least one of the $n$ predicted trajectories shares the same style label as the ground-truth, the sample is marked a *Hit*; otherwise it is a *Miss*. This hit/miss outcome directly measures whether the model's multi-modal outputs *cover* the driver's actual style, beyond conventional displacement errors.

## E  STYLE CONSISTENCY METRIC

To explicitly measure a model's ability to cover the correct driving style, as shown in Figure 6, we propose the *Style Miss Rate* a style consistency metric based on kinematic clustering following the clustering methodology introduced in (Zheng et al., 2025), and a hit/miss criterion:

**Extract kinematic static:** For each trajectory $\tau$, compute a feature vector

$$\phi(\tau) = \left[\max_t |a(t)|,\ Var(a),\ Var(v),\ \gamma\right]^\mathsf{T} \in \mathbb{R}^d, \tag{8}$$

where $a$ is acceleration (with $\max_t |a(t)|$ denoting the peak absolute acceleration over the trajectory, i.e. the highest instantaneous acceleration magnitude), $v$ is speed, and

$$\gamma = \frac{\mathrm{Var}\big(j(t)\big)}{\mathbb{E}\big[j(t)\big]}$$

is the jerk-variance ratio as defined in Murphey et al. (2009).

**Learn style clusters:** Fit a Gaussian Mixture Model (GMM) with $k = 2$ on the set of all ground-truth kinematic embeddings in the evaluation set $\{\phi(\tau_n^*)\}_{n=1}^N$, yielding a cluster assignment function

$$C(\phi) \in \{\texttt{"normal"}, \texttt{"aggressive"}\}. \tag{9}$$

Normal and aggressive are assigned based on the mean speed of each cluster. A cluster with higher mean speed will be assigned as aggressive.

**Assign styles to predictions:** For each sample $n$, let $\{\hat{\tau}_{n,i}\}_{i=1}^6$ be the six predicted trajectories. Define

$$s_n^* = C\big(\phi(\tau_n^*)\big), \quad s_{n,i} = C\big(\phi(\hat{\tau}_{n,i})\big). \tag{10}$$

**Define hit/miss:** A *hit* occurs if at least one predicted style matches the ground-truth style:

$$\mathrm{Hit}_n = \{\exists i \ : \ s_{n,i} = s_n^*\}, \quad \mathrm{Miss}_n = 1 - Hit_n. \tag{11}$$

**Style Miss Rate:** The overall metric is

$$\mathrm{SMR} = \frac{1}{N} \sum_{n=1}^N \mathrm{Miss}_n. \tag{12}$$

By construction, the SMR goes beyond pure spatial accuracy: it measures whether the model "covers" the driver's true style among its multi-modal outputs. A style-agnostic predictor may achieve low ADE/FDE by clustering its modes around average behavior, but will incur a high miss rate on aggressive samples. In contrast, a style-aware model—conditioned on inferred driving-style embeddings—should include at least one candidate trajectory whose kinematics align with the true style, yielding a lower SMR.

# F    STYLE MISS RATE EVALUATION ON ABLATION VARIANTS

To further assess how each component of our mixture-of-experts framework contributes to style coverage, we compute the Style Miss Rate (SMR) on the same ablation variants presented in Table 4. That is, for each model variant—removing reconstruction, KL loss, entropy regularization, etc.—we evaluate how often none of its multi-modal predictions match the true driving style cluster. The resulting SMR values are reported in Table 6. This analysis shows that the ablations which most degrade traditional error metrics (e.g. reconstruction and KL removal) also incur the largest increases in SMR, indicating a direct link between component contributions and the model's ability to cover the driver's style.

Table 6: **Style Miss Rate (SMR) for each ablation variant.**

| Ours | SMR$\downarrow$ |
|---|---|
| +LinearRouter | 0.2246 |
| +Full Finetuning | 0.2033 |
| -Social Forces | 0.2229 |
| -Context Features | 0.2236 |
| -KL Loss | 0.2267 |
| -Reconstruction | 0.2263 |
| -Expert Entropy Loss | 0.2260 |

## G  HOW DOES THE MODEL REASON? TAKING A LOOK AT THE SALIENCY MAPS.

To gain insight into which input features the model attends when forecasting agent trajectories, we compute saliency maps by measuring the sensitivity of the most likely predicted trajectory with respect to each input feature. Formally, let $X = \{A_{\text{in}}, M_{\text{in}}\}$ denote the concatenation of historical object trajectories $A_{\text{in}}$ and map polylines $M_{\text{in}}$. If $\hat{p}_\tau$ is the probability of trajectory $\tau$, then let $\hat{\tau}^* = \arg\max_\tau \hat{p}_\tau$ be the most likely predicted trajectory after a forward pass from a model. Then, we compute the gradient

$$\nabla_X \hat{\tau}^* = \frac{\partial \hat{\tau}^*}{\partial X}$$

via back-propagation, and form the saliency map

$$S(X) = \log(\|\nabla_X \hat{\tau}^*\|_2 + 1).$$

We use logarithmic scaling above to better display nuances in smaller gradient magnitudes. For visualizing this saliency map, we render the map polylines and the agents' historical trajectories colored using $S(X)$ on the "jet" color scheme (dark blue to green to dark red). Warmer colors highlight map segments or agent trajectories that the model deems more important for predicting future trajectories. Likewise, lighter colors highlight areas of less importance. Each agent vehicle is also colored on the same scale based on the maximum saliency value of its historical trajectory. We generate this visualization for MTR+Actions, Ours+CAT, and Ours+MoV:

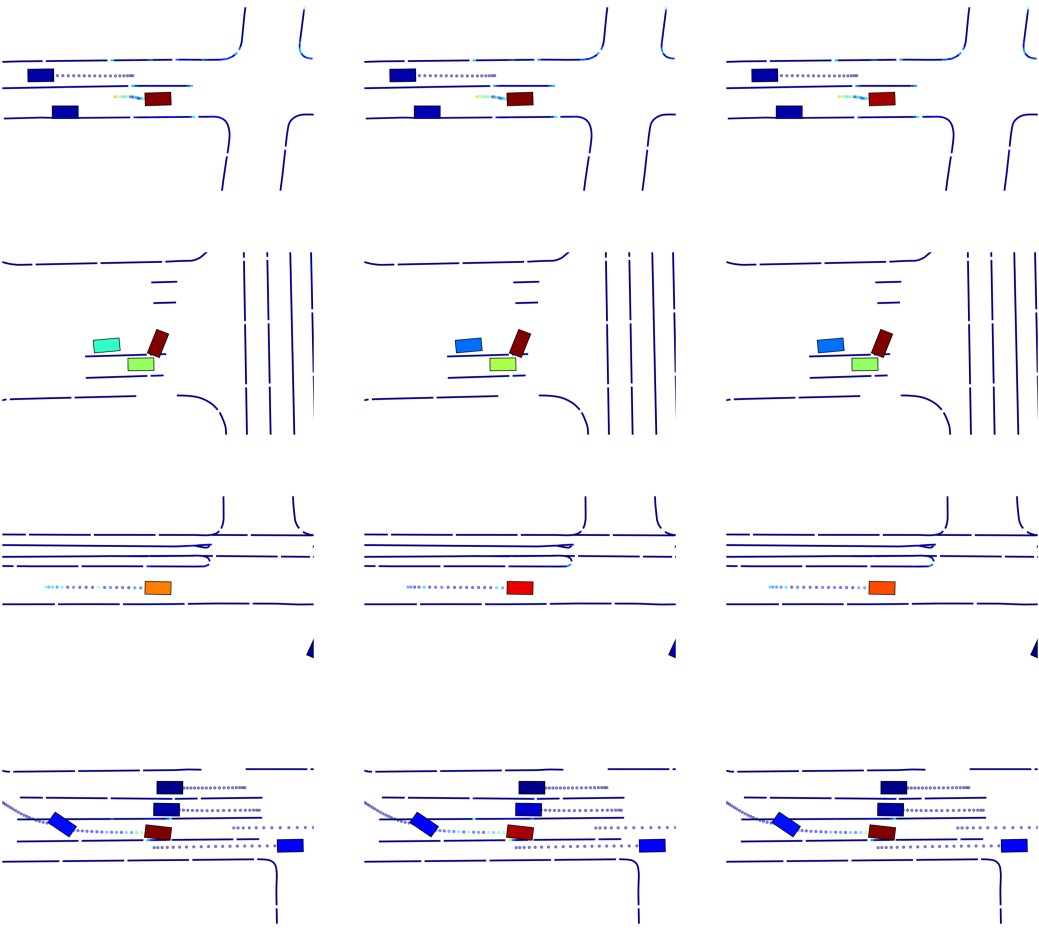

Figure 7: **Visualization of scenario feature saliency (Continued on the next page).** Saliency maps are visualized for 10 randomly selected scenarios from Argoverse on MTR+Actions (Left), Ours+CAT (Middle), and Ours+MoV (Right).

.

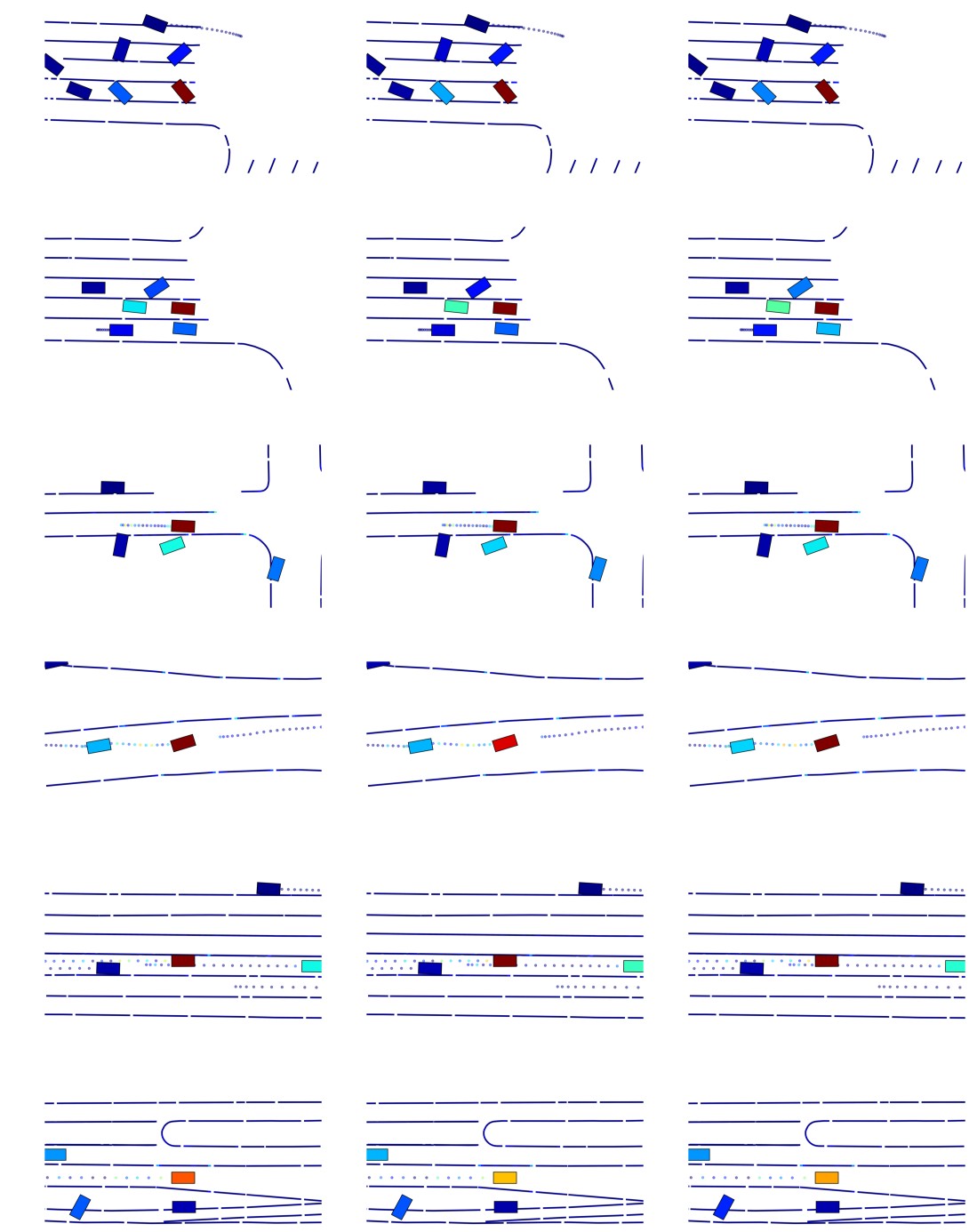

Figure 7: **Visualization of scenario feature saliency.** Saliency maps are visualized for 10 randomly selected scenarios from Argoverse on MTR+Actions (Left), Ours+CAT (Middle), and Ours+MoV (Right). The top 2 scenarios feature mostly stationary agents. In each scenario and model, the ego vehicle has the most saliency (colored red), followed by nearby agents and map polylines.

# H   ADDITIONAL VISUALIZATIONS FOR RELATIVE KINEMATICS BY EXPERT

(a) C-Poly

(b) HyperFormer

(c) Polytropon

(d) MoV

(e) CAT

Figure 8: **Kinematic magnitude comparisons for all variants of our approach.** Experiments run with seed 0 are plotted.

# I ADDITIONAL T-SNE PLOTS FOR ROUTER EMBEDDINGS BY EXPERT

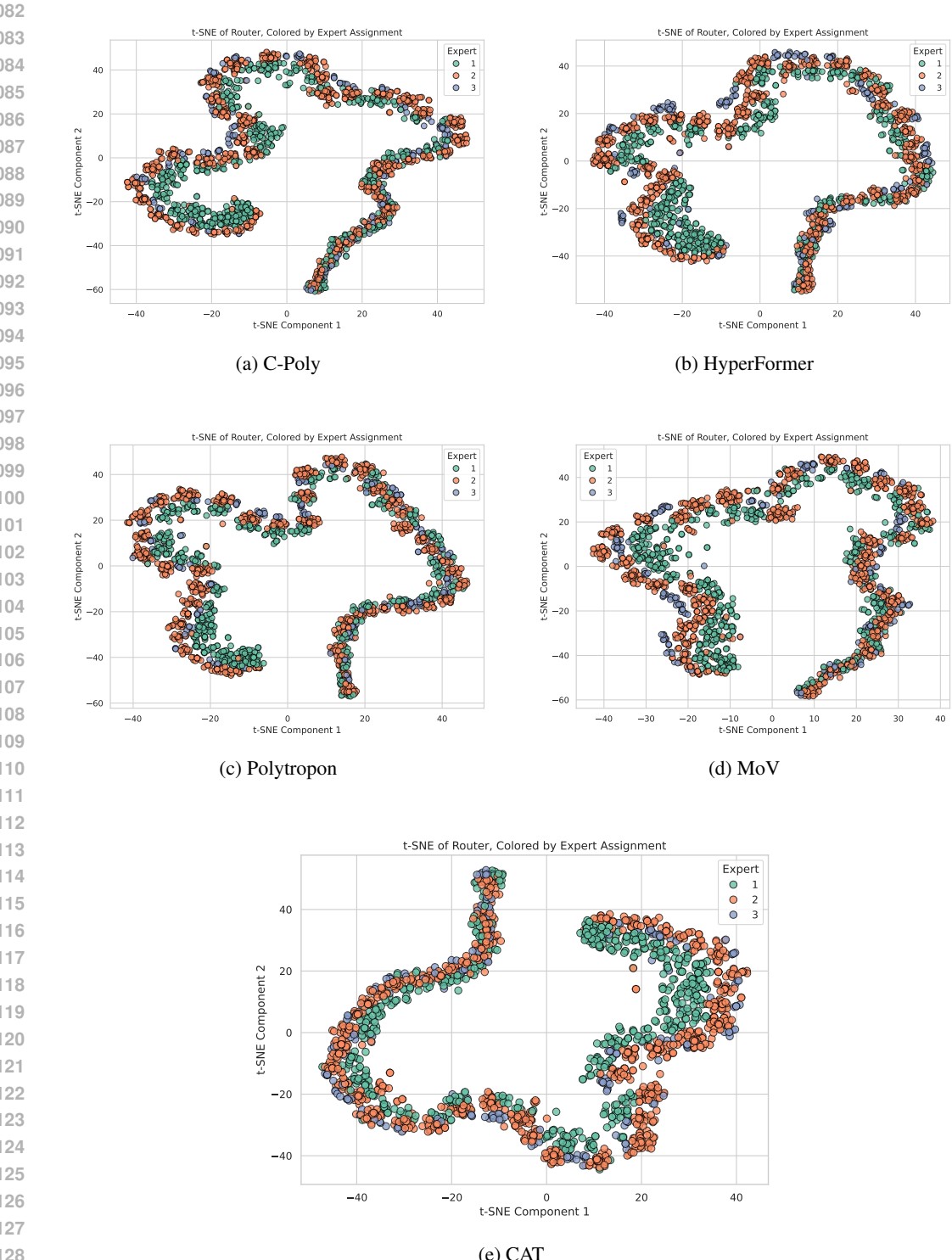

(a) C-Poly

(b) HyperFormer

(c) Polytropon

(d) MoV

(e) CAT

Figure 9: **t-SNE visualization of router embeddings for all variants of our approach.** Experiments run with seed 0 are plotted.

## J   BASELINE MTR MODEL HYPERPARAMETERS

Table 7: Hyperparameters used for training the MTR baseline model.

| Hyperparameter | Value |
| --- | --- |
| **Context Encoder** | |
| NAME | MTREncoder |
| NUM_OF_ATTN_NEIGHBORS | 7 |
| NUM_INPUT_ATTR_AGENT | 39 |
| NUM_INPUT_ATTR_MAP | 29 |
| NUM_CHANNEL_IN_MLP_AGENT | 256 |
| NUM_CHANNEL_IN_MLP_MAP | 64 |
| NUM_LAYER_IN_MLP_AGENT | 3 |
| NUM_LAYER_IN_MLP_MAP | 5 |
| NUM_LAYER_IN_PRE_MLP_MAP | 3 |
| D_MODEL | 256 |
| NUM_ATTN_LAYERS | 6 |
| NUM_ATTN_HEAD | 8 |
| DROPOUT_OF_ATTN | 0.1 |
| USE_LOCAL_ATTN | True |
| **Motion Decoder** | |
| NAME | MTRDecoder |
| NUM_MOTION_MODES | 6 |
| D_MODEL | 512 |
| NUM_DECODER_LAYERS | 6 |
| NUM_ATTN_HEAD | 8 |
| MAP_D_MODEL | 256 |
| DROPOUT_OF_ATTN | 0.1 |
| NUM_BASE_MAP_POLYLINES | 256 |
| NUM_WAYPOINT_MAP_POLYLINES | 128 |
| LOSS_WEIGHTS.cls | 1.0 |
| LOSS_WEIGHTS.reg | 1.0 |
| LOSS_WEIGHTS.vel | 0.5 |
| NMS_DIST_THRESH | 2.5 |
| **Training** | |
| max_epochs | 40 |
| learning_rate | 0.0001 |
| learning_rate_sched | [22, 24, 26, 28] |
| optimizer | AdamW |
| scheduler | lambdaLR |
| grad_clip_norm | 1000.0 |
| weight_decay | 0.01 |
| lr_decay | 0.5 |
| lr_clip | 0.000001 |
| WEIGHT_DECAY | 0.01 |
| train_batch_size | 64 |
| eval_batch_size | 64 |
| **Data** | |
| max_num_agents | 64 |
| map_range | 100 |
| max_num_roads | 768 |
| max_points_per_lane | 20 |
| manually_split_lane | True |
| point_sampled_interval | 1 |
| num_points_each_polyline | 20 |
| vector_break_dist_thresh | 1.0 |
| predict_actions | True |

# K PolySona Model Hyperparameters

Table 8: Hyperparameters used for training the PolySona models. Rows highlighted in yellow indicate differences from the baseline MTR configuration.

| Hyperparameter | Value |
|---|---|
| **Context Encoder** | |
| NAME | MTREncoder |
| NUM_OF_ATTN_NEIGHBORS | 7 |
| NUM_INPUT_ATTR_AGENT | 39 |
| NUM_INPUT_ATTR_MAP | 29 |
| NUM_CHANNEL_IN_MLP_AGENT | 256 |
| NUM_CHANNEL_IN_MLP_MAP | 64 |
| NUM_LAYER_IN_MLP_AGENT | 3 |
| NUM_LAYER_IN_MLP_MAP | 5 |
| NUM_LAYER_IN_PRE_MLP_MAP | 3 |
| D_MODEL | 256 |
| NUM_ATTN_LAYERS | 6 |
| NUM_ATTN_HEAD | 8 |
| DROPOUT_OF_ATTN | 0.1 |
| USE_LOCAL_ATTN | True |
| **Motion Decoder** | |
| NAME | PolySonaDecoder |
| NUM_MOTION_MODES | 6 |
| INTENTION_POINTS_FILE | `cluster_64_center_dict_6s.pkl` |
| D_MODEL | 512 |
| NUM_DECODER_LAYERS | 6 |
| NUM_ATTN_HEAD | 8 |
| MAP_D_MODEL | 256 |
| DROPOUT_OF_ATTN | 0.1 |
| NUM_BASE_MAP_POLYLINES | 256 |
| NUM_WAYPOINT_MAP_POLYLINES | 128 |
| LOSS_WEIGHTS.cls | 1.0 |
| LOSS_WEIGHTS.reg | 1.0 |
| LOSS_WEIGHTS.vel | 0.5 |
| NMS_DIST_THRESH | 1.0 |
| **Training** | |
| max_epochs | 10 |
| learning_rate | 0.001 |
| learning_rate_sched | [22, 24, 26, 28] |
| optimizer | AdamW |
| scheduler | polynomialLR (power=2) |
| grad_clip_norm | 1000.0 |
| weight_decay | 0.00 |
| lr_decay | 0.5 |
| lr_clip | 0.000001 |
| train_batch_size | 256 |
| eval_batch_size | 256 |
| predict_actions | True |
| lora_rank | 4 |
| freeze_encoder | True |
| freeze_decoder | True |
| attention_only | False |
| num_personas | 3 |
| prior | [0.3, 0.6, 0.1] |
| $\lambda_{recon}$ | 50 |
| $\lambda_{KL}$ | 50 |
| $\lambda_{entropy}$ | 25 |
| seed | 0 / 1 / 2 |
| **Data** | |
| max_num_agents | 64 |
| map_range | 100 |
| max_num_roads | 768 |
| max_points_per_lane | 20 |
| manually_split_lane | True |
| point_sampled_interval | 1 |
| num_points_each_polyline | 20 |
| vector_break_dist_thresh | 1.0 |

## L  IMPACT OF RANK ON PERFORMANCE

Table 9: **Comparison of Ours+CAT Across Different Ranks.**

| Rank | brierFDE↓ | minADE↓ | minFDE↓ | MissRate↓ |
|------|-----------|---------|---------|-----------|
| 2    | 2.1593    | 0.8573  | 1.7059  | 0.3151    |
| 4    | 2.1607    | 0.8624  | 1.7041  | 0.3171    |
| 8    | 2.1578    | 0.8578  | 1.7042  | 0.3120    |
| 16   | 2.1668    | 0.8610  | 1.7102  | 0.3151    |

Table 10: **Comparison of Ours+CAT Across Different Ranks, Grouped by Kalman Difficulty and TDBM Driving Styles.**

| Rank | Kalman Difficulty | | | TDBM Driving Styles | | | |
|------|------|--------|------|-------|---------|----------|-------------|
|      | Easy | Medium | Hard | Timid | Careful | Reckless | Threatening |
| 2    | 0.8120 | 1.1675 | 3.9875 | 0.8903 | 0.8865 | 0.8577 | 0.8172 |
| 4    | 0.8188 | 1.1678 | 2.4978 | 0.8793 | 0.8477 | 0.8655 | 0.8161 |
| 8    | 0.8130 | 1.1641 | 3.9882 | 0.8913 | 0.9833 | 0.8581 | 0.8183 |
| 16   | 0.8149 | 1.1758 | 4.2696 | 0.8944 | 0.8858 | 0.8613 | 0.8225 |

## M    STANDARD DEVIATION TABLE

Table 11: **Standard Deviation of Trajectory Prediction Benchmark Performance Comparisons.**

| Method | brierFDE↓ | minADE↓ | minFDE↓ | MissRate↓ |
|---|---|---|---|---|
| Ours+Polytropon Ponti et al. (2023) | 0.0004 | 0.0004 | 0.0004 | 0.0009 |
| Ours+C-Poly Wang et al. (2024a) | 0.0026 | 0.0011 | 0.0025 | 0.0003 |
| Ours+HyperFormer Karimi Mahabadi et al. (2021) | 0.0017 | 0.0004 | 0.0017 | 0.0010 |
| Ours+CAT Prabhakar et al. (2024) | 0.0019 | 0.0012 | 0.0018 | 0.0010 |
| Ours+MoV Zadouri et al. (2024) | 0.0010 | 0.0007 | 0.0010 | 0.0004 |

Table 12: **Standard Deviation of minADE Comparison by Kalman Difficulty and TDBM Driving Style groups.**

| Method | Kalman Difficulty | | | TDBM Driving Styles | | | |
|---|---|---|---|---|---|---|---|
| | Easy | Medium | Hard | Timid | Careful | Reckless | Threatening |
| Ours+PolyTropon Zadouri et al. (2024) | 0.0003 | 0.0032 | 0.0040 | 0.0005 | 0.0038 | 0.0004 | 0.0005 |
| Ours+C-Poly Wang et al. (2024a) | 0.0013 | 0.0003 | 0.0146 | 0.0009 | 0.0286 | 0.0012 | 0.0011 |
| Ours+HyperFormer Karimi Mahabadi et al. (2021) | 0.0006 | 0.0013 | 0.0060 | 0.0006 | 0.0330 | 0.0004 | 0.0007 |
| Ours+CAT Prabhakar et al. (2024) | 0.0012 | 0.0051 | 0.0682 | 0.0014 | 0.0340 | 0.0012 | 0.0012 |
| Ours+MoV Zadouri et al. (2024) | 0.0007 | 0.0006 | 0.0041 | 0.0008 | 0.0026 | 0.0007 | 0.0006 |

Table 13: Hyperparameter sweep of # of classes used in PolySona training.

| # Latent Classes | Easy / minADE | Medium / minADE | Hard / minADE | Overall minADE |
|---|---|---|---|---|
| 2 | **0.8108** | 1.1951 | 3.5275 | **0.8589** |
| 3 | 0.8188 | **1.1678** | **2.4978** | 0.8624 |
| 4 | 0.8119 | 1.1990 | 3.5116 | 0.8603 |
| 5 | 0.8120 | 1.1996 | 3.5191 | 0.8605 |
| 6 | 0.8109 | 1.1986 | 3.5729 | 0.8595 |
| 8 | 0.8123 | 1.2027 | 3.4217 | 0.8610 |
| 10 | 0.8122 | 1.2007 | 3.6048 | 0.8609 |

## N  HYPERPARAMETER SWEET: NUMBER OF LATENT CLASSES MODELED

