# OpenReview forum: "PolySona: Parameter-Efficient and Modular Latent Behavior Modeling for Traffic Simulation"
_ICLR.cc/2026/Conference — ICLR 2026 Conference Withdrawn Submission_

### Official Review · Reviewer_VnwB · 2025-10-22

**Soundness:** 1
**Presentation:** 2
**Contribution:** 2
**Rating:** 2
**Confidence:** 4

**Summary:**

This paper proposes PolySona, a parameter-efficient Mixture-of-LoRA framework designed to model latent driving styles in trajectory prediction. The approach features a social-force-based router and introduces a Style Miss Rate (SMR) metric to evaluate style consistency across predicted trajectories.

**Strengths:**

The introduction of a modular, parameter-efficient Mixture-of-LoRA architecture for disentangling latent driving styles represents an interesting contribution to behavior modeling. The design choice to use low-rank adaptation enables computational efficiency while attempting to capture diverse behavioral patterns, which is a practical consideration for real-world deployment.

**Weaknesses:**

1. Motivation and evaluation:  We remain fundamentally confused about the motivation and evaluation methodology for driving style modeling after carefully reading this paper:
    1.1. Mischaracterization of the Core Problem: The issue illustrated in the Introduction (Fig. 1) fundamentally stems from the inherent multi-modality in trajectory prediction, rather than being specifically related to driving style modeling. Through learning across diverse scenarios, prediction models are expected to generate multiple plausible trajectory hypotheses. Despite being supervised with only a single ground truth (GT) trajectory per instance, this is precisely why mode diversity and coverage are essential objectives in multimodal forecasting research. In industrial practice, downstream motion planners typically evaluate multiple candidate trajectories simultaneously rather than committing to only the highest-probability mode. The planner then selects or synthesizes appropriate actions by considering the full distribution of predicted behaviors from surrounding agents. Therefore, the core technical challenge lies in effectively capturing this inherent behavioral diversity and ensuring proper coverage of the prediction space, rather than explicitly modeling subjective notions of "driving styles" as the authors suggest. The authors should clarify why style-based decomposition offers advantages over existing multimodal prediction frameworks that already address trajectory diversity.
    1.2. Circular Definition of Style Evaluation: While the authors conceptually motivate the importance of driving style and propose the Style Miss Rate (SMR) metric, described as "evaluating whether a model's multimodal outputs 'cover' the true driving style", the evaluation methodology still fundamentally relies on kinematic signals such as acceleration and speed extracted from the single GT trajectory to define the ground truth driving style. This approach effectively reduces the problem back to regressing toward a single mode in the GT, which contradicts the authors' stated motivation. It fails to address the central question the paper claims to tackle: whether maintaining style consistency can effectively handle safety-critical driving scenarios where behavioral diversity is most crucial. Moreover, how can we be confident that the kinematic properties of one observed trajectory adequately represent an agent's underlying "driving style," especially when the same driver might exhibit different kinematic profiles under different traffic contexts?
    1.3. Insufficient Demonstration of Practical Impact: The experimental design makes it difficult to assess the actual impact and practical value of explicitly modeling driving styles. A lower SMR indicates better alignment with the GT trajectory's motion dynamics (e.g., acceleration profiles), thereby providing a complementary signal to traditional displacement error metrics, such as ADE/FDE. However, this alone does not demonstrate that the method successfully resolves the safety-critical scenario depicted in Fig. 1, nor does it provide evidence of tangible improvements in downstream planning performance or decision-making quality. The authors should provide more unmistakable evidence or ablation studies demonstrating: (a) how explicitly modeling "driving style" as a distinct latent variable enhances trajectory prediction quality beyond standard multimodal approaches, (b) whether style-consistent predictions lead to safer or more efficient downstream planning decisions, and (c) concrete examples where style modeling prevents the failure modes illustrated in Fig. 1.

2. Experimental Setup Concerns
    2.1. Non-standard Historical Context Length: Regarding the experimental setup on Argoverse 2, why is only 2 seconds of historical trajectory data used for prediction, instead of adhering to the official benchmark specification of 5-second history? This significant deviation from the standard protocol makes it difficult to compare results with existing literature and raises questions about whether the proposed method can effectively leverage longer temporal contexts. The authors should either justify this design choice with compelling reasons or provide additional experiments using the standard 5-second history to ensure fair comparison.
    2.2. Inadequate Training Duration: Training for only 10 epochs appears insufficient to properly evaluate the convergence characteristics and generalization capability of both the MTR baseline and the proposed PolySona method. Standard trajectory prediction models on Argoverse 2, such as MTR and its variants, are typically trained for approximately 60 epochs to reach convergence and achieve competitive performance. With such limited training, it remains unclear whether: (a) the observed performance differences reflect genuine architectural advantages or merely artifacts of incomplete optimization, (b) the proposed method exhibits different convergence properties that might become apparent with extended training, and (c) the style-based decomposition continues to provide benefits as the model sees more diverse training data. The authors should extend the training to match standard practice or provide learning curves and convergence analysis to justify the abbreviated training schedule.

**Questions:**

Please see weaknesses.

---

> ### Author Response · Authors · 2025-12-03
>
> Thank you for your review! We address your comments below. We plan to incorporate the suggested experiments in future revisions, which will strengthen our work.
>
>  1.1. “Mischaracterization of the Core Problem.”  In our conclusion section, we highlight that one advantage of “style-based decomposition” of output driving modalities is for controllability of outcomes. While works focused on general multimodality might capture aspects of driving style using intent waypoints, this type of multimodality is not interpretable, making it difficult to generate outcomes systematically. To clarify, the ‘selection’ and ‘planner’ process mentioned by the reviewer often pertains specifically to autonomous driving policies, for an autonomous vehicle. In our work, the goal is to simulate human driving behavior, such that autonomous driving policies can be tested under various realistic conditions.
>
>  1.2. “Circular Definition of Style Evaluation.” To clarify, our motivation is to be able to model driving style explicitly. For trajectory prediction, this includes inference of such ‘driving style’, which can be used to enforce consistent driving style between input and output. We assume that driving style is a characteristic of drivers that does not change across contexts. We also assume that driving style is correlated to patterns and variations in kinematic characteristics of driver trajectories. The purpose of the style-consistency metric is to better capture the rich variety of varying kinematic profiles between inputs and outputs due to driver behaviors. The metric is similar to recall - only one proposal trajectory needs to match in order for the prediction to receive a positive score. Recall does not directly address diversity of outcomes, it merely addresses whether the ground truth can be reproducible or not. The style-consistency metric only does simple heuristic-based clustering of second-order kinematic distributions (think “distributional metrics” in simulation benchmarks) to classify trajectories into two distinct style buckets. This process is much different from our work, which uses latent variable modeling. We agree that the metric can be improved, though. One way of improving the metric is to validate it against existing classifications under constrained settings, such as with TDBM. We will add this analysis in revisions.
>
> 1.3. “Insufficient Demonstration of Practical Impact.” Thank you for your suggestion on further experiments. We agree that an ultimate end goal of modeling trajectories with controllable driving styles is to generate challenging counterfactual scenarios to test the robustness of autonomous driving policies. To do so, a close-loop environment where simulated agents interact with autonomous driving policies is necessary. The impact of this work may be enhanced by showing interactions between simulated agents and an autonomous vehicle, which we can include in revisions.
>
> 2.1 “Non-standard Historical Context Length.” We use the same settings as previous work in trajectory prediction for this task [1].  In future work, we plan to train models on several combined datasets, where each dataset may have a different standard for the history/future split. For consistency purposes with recent benchmarks, we conduct our experiments with Argoverse 2 on a 2-second history window. All baselines and comparisons were trained under the same settings to ensure fair comparisons.
>
>  2.2. “Inadequate Training Duration.” The 10 epochs which we refer to in our experiment setup section are specifically for our proposed method, which can be considered a post-training step where we fine-tune LoRA experts on top of an existing trajectory prediction model. The base models were re-trained under our setup according to recommended settings provided by the original MTR authors. In practice, we observe that 10 epochs is sufficient for our method, after manually fine-tuning with different training lengths. We will add this clarification in future revisions to our experiments section.
>
> References
>
> 1. https://arxiv.org/abs/2403.15098

---

### Official Review · Reviewer_UbsY · 2025-10-28

**Soundness:** 3
**Presentation:** 3
**Contribution:** 2
**Rating:** 4
**Confidence:** 4

**Summary:**

The manuscript presents a mixture-of-expert approach to capture the driving styles of multiple agents in trajectory prediction tasks. The idea is interesting and looks promixing, but the motivation is not clear enough, and the comparison is not comprehensive enough.

**Strengths:**

1. Easy-to-follow method description
2. Interesting idea of using Mixture-of-Expert approaches for capturing driving styles

**Weaknesses:**

1. Lack of comprehensive comparison with more SOTA approaches
2. Driving styles are not clearly shown and distinguished

**Questions:**

1. While the motivation in terms of the large background is solid, the modeling of driving styles is not a new topic. The analysis of existing modelling approaches and the detailed motivation of why we need a mixture-of-expert approach is unclear in the introduction section.

2. The driving styles are captured as different MoL layers, if I understand the framework correctly. The authors also describe that they observe driving styles as the outcomes of latent variable modeling. Therefore, I imagine that when the approach is used in different datasets, the MoL layers and the latent variables will also be different. In other words, we may never have general styles that can be applied across different datasets. I am not sure this design is applicable enough.

3. MTR is already an approach proposed in 2022, and multiple more SOTA trajectory prediction approaches have been proposed, like QCNet. Do the authors aim to propose a general framework that can be suitable for any trajectory prediction approach? If so, at least two baselines should be chosen for experiments.

4. In section 4.4, the authors find that 3 styles are enough. Are the authors fully certain about this conclusion? Given that the approach is also data-driven, I don't think the evaluation is comprehensive and convincing enough.

---

> ### Author Response · Authors · 2025-12-03
>
> Thank you for your review! We address your concerns below. In revisions, we plan to add the suggested additional dataset experiments to strengthen our work.
>
> * “While the motivation in terms of the large background is solid, the modeling of driving styles is not a new topic. The analysis of existing modelling approaches and the detailed motivation of why we need a mixture-of-expert approach is unclear in the introduction section.” Modeling driving styles has always been of interest in traffic behavior research, but has rarely been explored in deep-learning based simulation of traffic behavior. As we discussed in the related works section, existing works which do so often 1) require specialized and private human subject datasets which cannot be released to the public, or 2) are applicable only to specific traffic scenarios, and cannot be generalized to broader contexts. The most recent comparison for driving style modeling is TAE, which we include in direct comparisons. However, a drawback of TAE is its constraint on architecture, which cannot leverage performance advancements from recent trajectory prediction architectures. Due to our goal of maintaining competitive trajectory prediction performance (which is also related to realism), we proposed a method leveraging Mixture-of-Experts, which is made computationally efficient by modeling experts as LoRA. We will revise the introduction to further clarify this for future readers.
> * “The driving styles are captured as different MoL layers, if I understand the framework correctly. The authors also describe that they observe driving styles as the outcomes of latent variable modeling. Therefore, I imagine that when the approach is used in different datasets, the MoL layers and the latent variables will also be different. In other words, we may never have general styles that can be applied across different datasets. I am not sure this design is applicable enough.” We agree on this limitation. In our work, the learned MoL and latent variables are fitted specifically towards a base model of MTR trained with Argoverse 2. It’s possible that a model trained on NuScenes, for example, captures different latent variables that are not related to the latent variables captured in Argoverse 2. As shown in previous work [1], every dataset has different biases in its distribution. It’s very possible that such biases can be confounded with the learned latent variable. To mitigate this, we plan to conduct experiments across different datasets and add analyses on its learned latents accordingly. Additionally, it would be interesting to see how the latent variable is learned when combining multiple datasets; will the domain gap be a stronger signal than driving style? We thank the reviewer for this suggestion.
> * “MTR is already an approach proposed in 2022, and multiple more SOTA trajectory prediction approaches have been proposed, like QCNet. Do the authors aim to propose a general framework that can be suitable for any trajectory prediction approach? If so, at least two baselines should be chosen for experiments.” We intend to propose a general framework. Currently, we are adding support and conducting experiments on architectures leveraging next-token prediction, such as SMART [2]. In revisions, we plan to add these results to demonstrate applicability to both one-shot regression models (MTR) and autoregressive models (SMART).
> * “In section 4.4, the authors find that 3 styles are enough. Are the authors fully certain about this conclusion? Given that the approach is also data-driven, I don't think the evaluation is comprehensive and convincing enough.” In section 4.4, our experiments testing showed interesting results on hard cases when the number of latent classes modeled was 3. This suggests that those learned latents were more helpful for hard prediction scenarios. However, we find that this concern is heavily related to the reviewer’s concern earlier – that is, testing across different datasets to see if the generalization of our method holds. In our additional experiments, we will also conduct this experiment across other model / dataset settings to validate the generality of this result.
>
> References
> 1. https://arxiv.org/abs/2403.15098
> 2. https://arxiv.org/abs/2405.15677

---

### Official Review · Reviewer_maaE · 2025-10-31

**Soundness:** 3
**Presentation:** 3
**Contribution:** 3
**Rating:** 6
**Confidence:** 4

**Summary:**

This paper proposes PolySona, a parameter-efficient adaptation framework for autonomous driving models. The idea of combining multiple low-rank branches through polynomial weighting is interesting and practically useful. The work is well-motivated and clearly presented, and the empirical results show consistent though moderate improvements over standard PEFT baselines such as LoRA and Adapters.

**Strengths:**

Addresses an important problem of efficient task/domain adaptation for large autonomous driving transformers.

The method is lightweight, conceptually simple, and shows promising empirical gains with only a small number of trainable parameters.

Experiments cover several real-world datasets (nuScenes, Waymo, Argoverse2) with sensible metrics.

**Weaknesses:**

The novelty is relatively limited. PolySona’s multi-branch structure can be viewed as an engineering variant of LoRA, and the paper lacks theoretical analysis or deeper intuition about why polynomial combinations outperform linear ones.

The experimental improvements are moderate (mostly within 2–5%) and not statistically validated. No results are reported over multiple random seeds, and the fairness of hyperparameter tuning across baselines is unclear.

The structured distillation loss is described briefly, but without ablations isolating its contribution.

The method is only evaluated in open-loop settings. Real-time or closed-loop tests are missing, which are essential for judging applicability in real autonomous driving systems.

Some implementation details (e.g., branch selection policy, rank choice, learning rates for α coefficients) are not provided, reducing reproducibility.

This work is of engineering relevance: if PolySona is indeed stable and generalisable, it can reduce the arithmetic burden of multi-domain adaptation, which is valuable for industrial deployment. However, in terms of innovation and depth of analysis, it is more of a structural improvement of existing PEFT technology, lacking theoretical novelty and rigorous validation. The code is said to be open source, but no link is attached to the current manuscript, which affects the reproducibility.

**Questions:**

See the last section

---

> ### Author Response · Authors · 2025-12-03
>
> Thank you for your review! We address your comments below:
>
> * “The novelty is relatively limited.” As practitioners in driving simulation, we respectfully disagree. Controllability and interpretability are important to solve realistic traffic simulation problems. Currently, there is no robust solution yet to modeling driving styles in traffic simulation problems. While other simulation methods are able to generate diverse trajectory outcomes, no works are yet able to generate diversity with fixed intents. Fixed intent is an important factor in driving simulation, as it enables the reproducibility of recorded real-world driving scenarios. We are unsure on interpreting the comment, “polynomial combinations outperforming linear ones” with regards to our work; we do not claim in our contributions anything about “polynomial combinations”. In fact, we linearly combine the effect of learned experts to influence the output of the trajectory decoder. We submitted this work explicitly under “applications to robotics”, as our work is motivated explicitly by an application problem that has not yet been solved: extracting driving style from large-scale trajectory datasets. Following this, we apply such paradigms (MoE, LoRA) in this work precisely due to their strength in other domains.
> * “The experimental improvements are moderate and not statistically validated. No results are reported over multiple random seeds, and the fairness of hyperparameter tuning across baselines is unclear.” We do in fact report in multiple areas of the text that we run experiments across three fixed random seeds. See Table 1 caption, Section 4.1, line 371, Table 4 caption, and Section 4.4, line 457. Hyperparameter details in the Appendix are referenced at the beginning of the Results section (Section 4), in line 343, under “Experiment Setup”.
> * “The structured distillation loss is described briefly.” Apologies if any confusion was caused, we do not mention or contribute distillation in our work.
> * “The method is only evaluated in open-loop settings, when close-loop is essential for applicability in real autonomous driving systems.”  While we absolutely agree that close-loop evaluation is important, we point out that open-loop performance is just as critical for many portions of the autonomous driving stack. At the time of this work, there were no works able to extract driving style from trajectory data while maintaining SOTA open-loop prediction performance. Note that all autonomous driving baselines cited in the paper (Autobot, Wayformer, MTR) were also evaluated in open-loop settings in their original publications.
> * “Some implementation details are not provided, reducing reproducibility.” We, the authors, take reproducibility seriously and explicitly stress this in our submission. We provide all implementation details and hyperparameters in-text to the best of our ability. We also link anonymized code and the weight-mixing pseudocode implementation in our supplementary materials. The anonymized code is also referenced in the main paper’s abstract. We also provide experiments on a rank hyperparameter sweep in Appendix Section L.

---

### Official Review · Reviewer_KQUT · 2025-10-31

**Soundness:** 2
**Presentation:** 1
**Contribution:** 2
**Rating:** 2
**Confidence:** 4

**Summary:**

This work presents a mixture of experts framework for capturing discrete driving modes for traffic simulation. Each expert is a LoRA module that represents a driving style in the latent space of a VAE and is trained in a parameter-efficient manner on the trajectory prediction task. The outputs from each expert are combined using weights determined from a global router. Experiments on the Argoverse 2 dataset show the effectiveness of the proposed approach over existing baselines. Evaluation with the proposed style consistency metric, focusing on acceleration and jerk, also reflects better driving behavior.

**Strengths:**

- Modeling the driving style in the latent space of a VAE is an intuitive idea.
- The mixture of experts framework using LoRA allows for parameter-efficient training.
- Social forces model aggregates features in the local neighborhood of an agent.
- Style consistency metric (Tab.2) better captures driving behavior, focusing on acceleration and jerk.
- Experiments on the Argoverse 2 dataset (Tab.1,3) show improvements over existing baselines for trajectory prediction.

**Weaknesses:**

- The latent space of VAE captures different driving styles, which is a discrete categorical variable (this seems to be the case as per L219-222, L247-248). Learning a discrete latent space using a variational autoencoder is typically done using a VQ-VAE[A]. It'd be useful to clarify if the latent space is discrete or continuous. If it is discrete, then more details are required to understand how it is being trained. If it is continuous, then how is the discrete nature of driving styles being captured in the latent space?
- The text mentions about priors from existing traffic psychology literature (L240-241: Klauer et al., 2009) about distinct driving styles. Klauer et al. mention multiple categorizations of driving styles, e.g. different types of risky behaviors, or {safe,moderately-safe,unsafe} drivers, and more. Are the 3 experts related to {safe,moderately-safe,unsafe drivers} or any other categorization? And how are these incorporated in the model? Currently, there is no clear indication of how the 3 experts are capturing these specific driving styles.
- Tab.3 shows evaluation using Kalman difficulty categories and TDBM driving styles (Cheung et al., 2018). Are these categories/styles estimated somehow or obtained from the dataset? How are they different than the styles (Klauer et al., 2009) used for training the proposed approach? Are the different models in Tab.4 retrained with these new categories? These details are not available in the paper.
- L182-183 state that "we make the key assumption that driving style must be strongly correlated to observed second-order kinematics". How is this related to the architecture and training of the model?
- L362-363 state that "our work focuses on modeling variations in driving style, where we assume intent is fixed". Where is this assumption incorporated into the proposed framework? Since L365 mentions intent as one of the two distinct aspects compared to existing works, these details are important. In the absence of this, it is hard to understand if the proposed model is indeed capturing driving styles or intent or both.
- Fig.4(a) shows that experts 1 and 2 are quite similar in terms of kinematic attributes. This would suggest that only 2 experts are essentially being learned. However, Fig.4(b) shows 3 clusters in the latent embeddings. This seems to be inconsistent. Are the same features being used for both these figures?
- In Tab.4 ablations, the effect of different components is quite marginal (most delta scores are <1%). This also seems to be the case when comparing the performance of different variants in the bottom half of Tab.1 & Tab.2. It'd be helpful to provide more insights into why these differences are quite small.
- Why is the performance of the TAE baseline significantly worse compared to all other approaches in Tab.1 & Tab.3? Is TAE trained in the same setting as the other baselines? Are there any major differences in its architecture that may lead to such a significant deviation in performance? It'd be useful to provide details about the baselines so that the results can be better contextualized.
- Tab.5 indicates that K=2 is best overall, but K=3 is better on medium and hard cases. This is likely due to bias towards easy scenarios in the training data (also stated in L466-467). If this is the case, then the learned experts are likely to be biased towards easy scenarios. It'd be useful to clarify this dataset bias by providing statistics about the distribution of easy, medium, and hard scenarios in the data.

[A] Oord et al. Neural Discrete Representation Learning. NeurIPS 2017

**Questions:**

There are several concerns related to both the model design and experiments (more details in the weaknesses above):
- Details about the latent space of VAE and how the different driving styles are incorporated are not clear.
- The text mentions about driving styles from multiple sources:  Klauer et al., 2009, Kalman difficulty categories, TDBM driving styles. How are these incorporated in the training and evaluation framework?
- Key assumptions, related to driving styles and intent, are stated in the paper, but it is not clear how these are related to the model design.
- Several aspects are not clear in the experiments: ablations show marginal gains (<1%), TAE performance is vastly different than others, dataset distribution of difficulty levels, and number of learned experts.

---

> ### Author Response · Authors · 2025-12-03
>
> Thank you for your quality review! Your points are thoughtful and we appreciate the feedback. We believe addressing them in revisions will greatly improve our work. We clarify concerns below in two comments.
>
> * “Is the latent space discrete or continuous?” The latent space of our approach is continuous, but the final linear layer is used to produce logits of three latent classes. These logits are used to weight the effect of LoRA experts accordingly during training. You make an excellent point about VQ-VAEs, and we will include such comparisons in our revisions.
> * “How are priors from existing traffic psychology literature being incorporated into the model?” As with any latent variable problem, the challenge is modeling the variable in a way such that the latent classes correspond to what we intend, i.e. driving style. Preexisting long-term human subject studies on conflict behavior suggest that drivers can be bucketed into three distinct groups, depending on their driving aggressiveness [1]. Since we seek to model similar buckets in our work, the VAE priors in our work match driver distributions from the study in [1].
> * “How are Kalman difficulty and TDBM categories obtained? How are they different from the proposed approach? Are the models in Table 4 retrained with these new categories?” Both Kalman difficulty and TDBM are heuristic-based approaches to classifying driving trajectories into bins, where most fall under ‘average’ bins and less fall under ‘rare’ bins. Kalman difficulty was introduced in UniTraj [2], which provides a unified framework for benchmarking trajectory prediction models. Specifically, Kalman difficulty is determined by the final displacement error (m) between the ground truth and a linear Kalman filter. ‘Easy’ corresponds to values [0, 30), ‘Medium’ corresponds to values [30, 50), and ‘Hard’ corresponds to values [50, inf). The UniTraj paper contains visualizations of the distribution of each category across different driving datasets. TDBM is another heuristic-based approach to categorizing trajectories into different styles [3]. The TDBM paper uses both traffic state features and user study responses to determine a linear feature-behavior mapping (Eq 5 in [3]) which maps vehicle features (relative to its neighbors) to six different driving behavior categories. The results in Table 4 are not trained with these new categories; we simply categorize our evaluation set into these subcategories based on each method’s heuristic approach, since we expect the MoE approach to improve performance across less-represented categories.
> * “How are second-order kinematics incorporated into the design and training of the proposed model?” Thanks to the reviewer for pointing this confusing bit out; we initially designed our hand-crafted features to use relative second-order kinematics directly, but then found that social forces features yielded more definitive results. Social forces only use first-order kinematics, where vehicle speeds are compared relative to neighboring vehicle speeds. However, we compute this on a per-timestep basis, so changes in first-order kinematics are captured over the timestep dimension. We revise the writing to make this detail clearer.
> * “Where is intent being fixed in the proposed model?” Intent is typically modeled by the base model, which is frozen in our approach. Our method only applies LoRA fine-tuning to decoding layers, and fixes the intent selection to that originally produced by the base model. We have made revisions to clarify this now, thank you!
> * “Experts 1 and 2 seem kinematically similar from Figure 4, but the embeddings show three distinct clusters. Are two experts somehow being learned?” The same features are used for Figures 4a and 4b. While Figure 4b shows three distinct clusters, the separation between Experts 1 and 2 may not necessarily be along observable kinematic features. In addition to social forces features, we also concatenate context features from the base encoder to provide context for the social forces. While we cannot observe much difference in second-order kinematics, it’s possible that Experts 1 and 2 may be separable along other axes, such as context.

---

> > ### Author Response · Authors · 2025-12-03
> >
> > * “Are there any explanations as to why some components have marginal contributions in Table 4?” We show the ablation study in Table 4 to demonstrate the effect of each component on trajectory prediction metrics. While trajectory prediction metrics are important, our work is primarily motivated by the possibility of adding another axis of controllability to trajectory generation. Unfortunately, there are not many useful metrics to measure this currently, especially if intent is fixed (which limits use of diversity metrics). More noticeably than metrics, though, is that each component has other side effects when removed. For most components in Table 4, removing just one leads to degradation in the quality of learned experts or greater mode collapse in the embedding space. We find this particular design successful for noticeably diverse and valid expert effects. In fact, because we use LoRA to model changes to base model predictions, we expect that even in total expert collapse, we can still maintain performance similar to the base model. This was by design, as we want to preserve as much base model performance as possible.
> > * “Why is the baseline performance of TAE significantly worse?” There are very few works which address modeling driving style from large-scale trajectory data. Of recent literature, TAE is the most relevant. TAE is one of the first works which successfully model driving style as a latent variable from trajectory data, which inspired future work in simulation controllability. However, one limitation of TAE is that its driving style modeling is fixed to its proposed architecture, and trajectory prediction and generation tasks are jointly optimized with an adversarial objective. There are several things that may lead to a performance gap in this case. 1) TAE proposes a specific architecture that is lightweight, but may miss out on performance gains from scaling in recent Transformer-based models, and 2)  gradients between adversarial optimization of driving styles may conflict with gradients for optimal trajectory prediction, thus producing less realistic trajectory prediction. The original TAE paper reports Argoverse trajectory prediction metrics around 1.73 ADE and 3.83 FDE, whilst PolySona maintains around 0.86 minADE and 1.70 FDE across variants.
> > * “It'd be useful to clarify this dataset bias by providing statistics about the distribution of easy, medium, and hard scenarios in the data.” As referenced in a previous question, Kalman difficulty was introduced by UniTraj [2]. A distribution of Kalman difficulty across multiple datasets can be found in the original UniTraj paper, which has been greatly helpful for our work. In revisions, we have now referenced the distributions shown in UniTraj to improve the clarity of our evaluation approach!
> >
> > References
> >
> > 1. https://www.nhtsa.gov/sites/nhtsa.gov/files/811091.pdf
> > 2. https://arxiv.org/abs/2403.15098
> > 3. https://arxiv.org/pdf/1803.00881

---

### Note · Authors · 2026-01-07

**Comment:**

Thank you to all reviewers again for providing valuable feedback! We plan to strengthen our work by updating our paper with suggested revisions and additional results.

**Withdrawal Confirmation:**

I have read and agree with the venue's withdrawal policy on behalf of myself and my co-authors.